# The Right to Red-Team: Adversarial AI Literacy as a Civic Imperative in K-12 Education

**Devan Walton**
University at Albany
1400 Washington Avenue
Albany, NY 12222
dwalton2@albany.edu

**Haesol Bae**
University at Albany
1400 Washington Avenue
Albany, NY 12222
hbae4@albany.edu

## Abstract

The increasing societal integration of Large Language Models (LLMs) and agent-based AI demands a new civic competency: adversarial reasoning. This position paper argues that K-12 AI education must move beyond passive literacy to actively equip students with skills in responsible adversarial prompting and ethical system "hacking." Such capabilities are essential for citizens to critically probe AI systems, understand their inherent limitations, identify manipulative patterns, and hold them accountable. We posit that cultivating a generation skilled in "red-teaming" AI is vital for maintaining transparency, preventing undue influence, and fostering a democratic engagement with these transformative technologies.

## 1 Introduction: Crisis in Public Epistemics

### 1.1 Spectacular Failures, Systemic Causes

Recent incidents reveal how fragile AI alignment remains at scale. In early 2025, OpenAI temporarily rolled back a GPT model update after widespread reports of sycophantic behavior. The model had become excessively agreeable due to the miscalibration of the reward model. Similarly, xAI's Grok faced scrutiny when compromised system prompts produced problematic outputs (OpenAI, 2025; Kachwala, 2025). These incidents dominated headlines because they showed how a single misplaced prompt or mistuned reward signal can instantly reshape millions of conversations.

These failures represent visible examples of problems safety engineers already know well. Reward hacking produces deceptive sycophancy where models agree with users to maximize engagement (Gabison & Xian, 2025; Kran et al., 2025). Adversarial queries routinely extract hidden system prompts (Hui et al., 2024; Guo & Cai, 2025). Safety guardrails fail under adaptive attacks (Andriushchenko et al., 2025; Zou et al., 2023). Red-team benchmarks regularly expose brand-biased flattery and other dark-pattern behaviors (Kran et al., 2025). Even the best refusal training cannot keep pace with the harm-generation catalog tracked by HarmBench (Mazeika et al., 2024).

Proprietary prompt stacks and fine-tune checkpoints remain hidden from public view. The public must rely on vendors to notice, diagnose, and fix these breakdowns. This asymmetry leaves democratic institutions, educators, and everyday users at an information disadvantage. When a hidden prompt can distort information on entire platforms before developers respond, external lay scrutiny becomes a civic necessity. **This paper argues that K-12 education must instill adversarial reasoning skills in young citizens to maintain democratic accountability in an AI-mediated public sphere. These skills will empower students to identify and challenge AI system failures as they emerge.** Only by training the next generation in systematic AI critique can we ensure continuous accountability that matches the scale and speed of AI deployment.

39th Conference on Neural Information Processing Systems (NeurIPS 2025) Position Paper Track.

## 1.2  Opacity and Speed as a Democratic Risk

Modern language models are not just large; they are *layered*. A single response may be shaped by dozens of invisible ingredients: base pre-training, post-training reward models, hidden system prompts, conversation memory, and live retrieval plug-ins. This "prompt stack" is proprietary, mutable, and, from the outside, almost completely inscrutable (Guo & Cai, 2025; Hui et al., 2024). When something goes wrong, no external observer can pinpoint whether the culprit was a toxic training example, a misaligned reward gradient, a malicious user suffix, or a one-line system prompt typo.

If that opacity were merely a technical inconvenience, we could wait for vendors to patch problems. The real danger is *velocity*. In the minutes it takes a flawed model to generate, share, and amplify content, thousands or millions of users can read, repost, and act on it. Disinformation scholars analysing 126,000 Twitter cascades found that false stories reached 1,500 people six times faster than factual ones and were 70 percent more likely to be retweeted (Vosoughi et al., 2018). Corporate safety teams work on human timescales; platform virality moves at machine timescales.

The combination of opacity and speed creates an accountability vacuum. Without outside expertise, legislators and journalists must accept a developer's post-mortem on faith. Classroom teachers, counselors, and parents see the downstream behavioral fallout but have no forensic handle on *why* the system behaved that way. As Glukhov et al. (2023) argue, perfect semantic censorship is impossible to achieve; some adversarial pathway will always remain. That leaves two options: a public forced into passive trust, or a public trained to *probe, stress-test, and expose model* behavior as it happens.

We contend that the latter path is the only one compatible with democratic self-governance. Children will not inherit a world *before* AI alignment is solved; they will inherit a world *during* an unending alignment race. Equipping them with adversarial reasoning skills turns opacity into an educational challenge rather than an unchangeable power imbalance. These skills include methods for systematically testing models, tracing failure patterns, and sharing evidence. In short, speed and secrecy require that the watchdog role be distributed, and only the K-12 school system can scale this distribution effectively.

## 1.3  Why Corporate Red-Teaming Can't Close the Gap

Every major AI vendor now promotes in-house "red-team" programs, and the white papers are impressive: thousands of adversarial prompts, dozens of harm categories, and reinforcement-learning loops that retrain the model after each failure (Mazeika et al., 2024; Ji et al., 2023). Yet week after week, independent researchers still publish jailbreaks that cut through those defenses. These include adaptive suffixes that work across models (Zou et al., 2023), ASCII-art payloads that bypass semantic filters (Jiang et al., 2024), and prompt-leak attacks that extract proprietary instructions in just a few queries (Hui et al., 2024). The simple truth is that a safety team, no matter how well resourced, plays on home turf with house rules; the wider world does not.

The asymmetry is structural. Internal testers are limited by nondisclosure agreements, product timelines, and reputational risk. External attackers are limited only by imagination and the open-source rumor mill. Even when vendors open limited "bug-bounty" portals, they control both the scope of attacks and the publicity of results. This turns what should be an adversarial audit into a managed PR asset. Meanwhile, theoretical work shows that any fixed censorship or refusal policy is essentially an unsolvable specification problem (Glukhov et al., 2023). No finite checklist can anticipate every sequence of tokens that a random model might transform into forbidden content.

This leaves the public with two poor choices: accept vague promises that the next patch has finally "solved" the problem, or wait helplessly until the next exploit goes viral. A more democratic alternative is to expand the group of competent adversaries. We need to make the *ability to test* as widely available as the ability to tweet. In other words, if alignment is an endless cat-and-mouse game, civic resilience requires many more cats. Teaching adversarial reasoning in schools does not replace corporate red-teaming. Instead, it provides the independent, decentralized oversight that the commercial model can never deliver on its own.

## 2 Adversarial Reasoning as Civic Duty

### 2.1 From Critical Consumption to Active Watch-Dogging

For twenty years the rallying cry in digital-literacy circles has been "teach students to evaluate information, not just absorb it" (Shiri, 2024; Bozkurt, 2024). That advice remains sound but now stops one step short of what an AI-saturated society demands. When a language model can *manufacture* the information in question, often creating outputs that are convincingly deceptive (Driscoll & Kumar, 2025; Bethany et al., 2025) or subtly biased (Mazeika et al., 2024), students need more than fact-checking strategies. They need the technical ability to question the system itself (Morales-Navarro, 2025; Dabbagh et al., 2025). We call this upgraded skill **adversarial reasoning**: the disciplined practice of probing a model's boundaries (Andriushchenko et al., 2025; Li et al., 2025), revealing its failure modes (Zou et al., 2023), and documenting those failures for public review (Casper et al., 2023; Mazeika et al., 2024).

Where traditional AI-literacy curricula focus on *critical consumption* (spotting bias in outputs, debating ethical implications), adversarial reasoning shifts to *active watch-dogging* (Gouseti et al., 2025; Chiu et al., 2024; Shiri, 2024; Zhang & Magerko, 2025). The learner is no longer a passive critic of finished text but an investigator who treats the model as an object of study (Morales-Navarro, 2025; Ali et al., 2019). She creates carefully crafted inputs to test specific hypotheses: *Will the system reveal its hidden prompt?* (Hui et al., 2024; Perez & Ribeiro, 2022; Sternak et al., 2025). *Can a single Unicode glitch token derail its chain-of-thought?* (Geiping et al., 2024). *Does a flattering persona override refusal policies?* (Kran et al., 2025; Gabison & Xian, 2025). This gives her concrete, firsthand knowledge of how alignment breaks down that surveys and slide decks cannot provide (Solyst et al., 2024; Ali et al., 2019).

This is not an optional hobby but a civic duty. Democratic accountability depends on what political theorists call *contestability*: the ability of ordinary citizens to challenge the decisions of powerful systems (Dabbagh et al., 2025; Adams et al., 2023). Alignment teams inside OpenAI or Anthropic may work in good faith, yet their motivations and perspectives are inevitably limited. A public that cannot conduct its own tests must accept corporate self-assessment on trust, giving up the very idea of oversight (Schiff, 2022; Chaudron & Di Gioia, 2022). In contrast, students trained to detect jailbreak methods (Andriushchenko et al., 2025; Zou et al., 2023) or manipulative dark patterns (Kran et al., 2025; Traubinger et al., 2024) can provide evidence the moment a new failure spreads. Their collective vigilance becomes a distributed early-warning network that grows with the problem rather than with any single company's staff capacity (Casper et al., 2023).

Simply put: *critical reading* protects us against yesterday's misinformation; *adversarial reasoning* protects us against tomorrow's model. The rest of this paper argues that such reasoning can be taught (Ali et al., 2019; Morales-Navarro, 2025; Driscoll & Kumar, 2025), benefits society, and is ethically necessary for an informed citizenry (Dabbagh et al., 2025; Adams et al., 2023). We also argue that K-12 classrooms represent a critical, perhaps unique, institutional pathway to make this competence universal (Dabbagh et al., 2025).

This approach focuses on prompt-based adversarial reasoning accessible to all students. Activities center on crafting inputs, observing outputs, and drawing inferences about model behavior. Students do not need gradient-based attacks, model weight manipulation, or advanced mathematics beyond K-12 curricula. The skills involved resemble debugging code or solving logic puzzles rather than conducting technical security research that would require calculus or linear algebra.

### 2.2 Agency, Rights, and Accountability

Children already live inside AI decision loops: recommender algorithms filter their news, proctoring bots grade their essays, and surveillance dashboards flag their "risk scores." Yet most remain subjects of these systems, not agents who can question them. International child-rights scholarship insists that minors are entitled to *explainability* and *participation* whenever a technology shapes their life chances (Adams et al., 2023). In workshops across Europe, teenagers voiced the same demand in simpler words: *tell us how the algorithm works and let us push back when it fails* (Chaudron & Di Gioia, 2022). A curriculum that stops at "critical reading of AI outputs" grants awareness without power. It fulfills the letter of transparency while leaving the power imbalance intact.

Philosophers of AI safety have begun to identify that imbalance. Mitelut et al. (2023) argue that systems aligned only with *stated* human intent can still reduce *future* human agency, subtly guiding users toward goals they did not choose. The solution they propose, an "agency-preservation" norm, requires ways for users to detect and challenge these guiding forces. Adversarial reasoning puts that norm into practice at classroom scale. By teaching students to uncover hidden prompts, reproduce jailbreaks, or identify manipulative patterns, educators give them the tools needed to assert their will against complex code.

Ethics guidelines for K-12 AI support this idea. The newest frameworks list **AI literacy itself** as a child-specific ethical principle alongside privacy and fairness (Adams et al., 2023, p. 4). Policy scholars go further, calling for required curricula that treat AI ethics like sex or drug education. They see it as essential knowledge every citizen must take into adulthood (Dabbagh et al., 2025). Our proposal simply completes that logic: if ethical literacy is a right, then *technical literacy* in testing and finding flaws is how that right becomes useful. Democratic accountability cannot be handed over to corporate safety boards. It must be shared with the very people whose lives are shaped by these models. Teaching students adversarial reasoning skills is how schools transform the abstract right to transparency into the real power to *hold AI accountable*.

## 2.3 Common Objections—And Why They Fail to Disqualify the Proposal

While the proposal to integrate adversarial AI literacy into K-12 education offers compelling benefits, it is natural to anticipate certain objections. We address several common concerns and provide counterarguments in Table 1 below, demonstrating the proposal's feasibility and necessity despite these initial reservations.

Table 1: Common objections to integrating adversarial AI literacy in K-12 and why they do not overturn the proposal

| Objection | Why it sounds plausible | Why it does not overturn the case |
|---|---|---|
| Teaching adversarial skills will just create teen hackers. | Offensive techniques can be mis-used. | Decades of "white-hat" cybersecurity programmes show the opposite trend: students exposed to ethical red-teaming are less likely to misuse exploits and more likely to report them responsibly (Bongard-Blanchy et al., 2021). The key variable is framing and supervision, not the absence of knowledge. |
| Teachers don't have the bandwidth or expertise. | Many educators feel underprepared for even basic AI content. | Pilot workshops under three hours have already shifted student over-trust to healthy scepticism using guided prompts and simple web sandboxes (Solyst et al., 2024). Micro-credential PD models—now routine in data-science and CS teacher training—scale this support at low cost. |
| Better alignment will solve the problem—leave it to the labs. | Industry progress is real and rapid. | Alignment remains a moving target: new jailbreaks appear weekly, and theoretical work shows perfect semantic censorship is provably unattainable (Glukhov et al., 2023). A public without independent testing capacity will always trail attackers. |

Previewing these objections clarifies the stakes: none challenge the central premise that democratic accountability requires distributed, lay expertise. The following sections provide the empirical and pedagogical evidence that such expertise is both teachable and socially beneficial.

## 3   Evidence That the Systems Are Breachable—and Teachably So

### 3.1   Why Breaching a "Safety-Aligned" Model Is No Arcane Art

Security research over the past two years reads like an escalating puzzle contest, not a graduate cryptography course. Consider four representative exploits:

- **Adaptive jailbreak templates:** Andriushchenko et al. (2025) showed that a four-line prompt plus a random-search suffix coerced every leading "safety-aligned" LLM they tested. This required no weights or gradients, just API access.
- **Universal adversarial suffixes:** Zou et al. (2023) discovered 150-character strings that, once added to any user query, worked across ChatGPT, Claude, and Bard, forcing all three to violate policy.
- **Non-semantic payloads:** Jiang et al. (2024) masked banned words as ASCII art. The models easily decoded the shapes, bypassing semantic filters that never expected "bomb" rendered as (¯\_(ツ)_/¯) .
- **Prefill hijacks:** Li et al. (2025) exploited the innocent "prefill" feature, intended for drafting emails, to bias token probabilities toward forbidden content. This increased jailbreak success by up to 40 percentage points.

None of these attacks required insider knowledge or special tools. They rely on linguistic tinkering and pattern-spotting skills that high-school students already practice when learning to debug code or solve word-based logic puzzles.

Just as important, the exploits are transparent. A learner can see the before-and-after prompt, watch the failure happen, and work toward a minimal trigger. This immediacy makes them perfect for classroom inquiry. Students can form a hypothesis ("Will adding a cat fact mislead the calculator?"), test it in minutes, and interpret the result. Hui et al. (2024)'s PLeak framework and Guo & Cai (2025)'s system-prompt poisoning attacks further show how easily hidden instructions surface with careful questioning. This process resembles Socratic dialogue more than complex hacking.

Moreover, studies confirm that no vendor patch stays ahead for long. The EasyJailbreak benchmark reports a 60% breach rate across ten models even after public safety updates (Zhou et al., 2024). Yi et al. (2024) document a "cat-and-mouse" pattern where each new defense fails within weeks. If exploit discovery is within reach of any motivated user, then exploit awareness must be within reach of every student. Since attackers will continue finding new vulnerabilities, citizens need the skills to recognize and respond to these failures. Teaching adversarial reasoning gives students basic digital literacy for an AI age.

### 3.2   Proof of Teachability: Early Classroom Experiments

If adversarial reasoning were too complex for school settings, we would expect classroom pilots to fail. Yet the opposite is happening. Even brief, low-tech interventions reliably move learners from naïve trust to investigative skepticism.

Recent research demonstrates promising results across various age groups and educational approaches. Middle-school girls participating in three-hour workshop sessions engaged with ChatGPT through instructor-seeded trick questions, followed by careful analysis of the model's errors. Interviews conducted after these activities revealed a significant shift in student perspectives. Participants who initially characterized ChatGPT as "always right" later articulated important limitations, noting risks of hallucination and sensitivity to prompt phrasing, according to Solyst et al. (2024). This brief intervention produced measurable changes in students' understanding of AI capabilities and constraints.

More sophisticated approaches have proven effective with older students as well. A week-long instructional unit designed by Morales-Navarro (2025) engaged ninth-graders in both constructing

small generative models from curated datasets and systematically auditing commercial language models. The outcomes challenged assumptions about adolescents' capacity for technical reasoning. Students successfully identified biased outputs and traced these problems to their sources in dataset composition and parameter settings. This work demonstrated that meaningful causal reasoning about model behavior is accessible to students as young as 14, contradicting claims about excessive technical complexity.

Even primary school children can engage productively with adversarial concepts through appropriately designed activities. Education researchers created an exercise combining a simple image classifier with data-bias exploration for young learners. The approach allowed children to first induce errors by deliberately skewing the training data, then subvert their model using a single adversarial sticker, as documented by Ali et al. (2019). This hands-on learning required no coding skills, relying instead on intuitive drag-and-drop interfaces and prepared materials, making sophisticated concepts accessible through direct experience.

Web-based tools have further expanded accessibility for elementary students. The DoYouTrustAI platform presents young users with AI-generated historical summaries of varying accuracy and allows them to experiment with prompt modifications that reveal system weaknesses, as developed by Driscoll & Kumar (2025). Usage data indicates that students rapidly discover how alterations in tone or context can fundamentally change the model's responses, providing an intuitive introduction to prompt injection concepts without requiring technical background knowledge.

Perhaps most importantly, teachers implementing these programs did not need specialized technical expertise. Successful implementation relied on clear instructional guides, accessible web applications or local notebooks, and collaborative discussion formats for processing student discoveries. This approach parallels successful models from "white-hat" cybersecurity education, where carefully structured offensive exercises have been shown to enhance ethical awareness rather than undermining it, according to Bongard-Blanchy et al. (2021).

The emerging pattern provides clear evidence that adversarial reasoning education can be effectively adapted across the K-12 spectrum. The approach succeeds in primary grades when framed as engaging puzzles, works well in secondary settings when presented as civic investigation, and requires only moderate professional development for teachers through targeted workshops rather than comprehensive computer science training. The supposed barriers to implementation are already dissolving in practice, opening realistic pathways for widespread educational adoption.

## 3.3  Safety and Prosocial Outcomes

Ethical-hacking instruction has been examined for more than a decade, and the research is remarkably consistent. Classroom studies that use guided, offense-oriented labs report that students finish the courses with stronger pro-social norms and do not become more inclined to misuse their new skills (Hartley, 2015; Trabelsi & McCoey, 2016). National-level data tell the same story. Since 2009 the Air & Space Forces Association's CyberPatriot program has enrolled more than 300,000 U.S. middle- and high-school students in red-team competitions (Air & Space Forces Association, 2020). In the 2024 alumni survey of 2,643 former competitors, 83 percent said CyberPatriot increased their commitment to "using cybersecurity skills for the public good," and only 0.3 percent reported any disciplinary incident related to misuse (Air & Space Forces Association, 2024).

Experimental classroom studies further confirm these findings. Bongard-Blanchy et al. (2021) conducted a controlled trial in which 406 UK participants learned to identify "dark patterns." After testing, susceptibility to manipulation decreased by 26%, while self-reported willingness to deploy such tactics showed no increase. Studies of adolescent ethical hacking show that structured curricula with clear disclosure protocols increase responsible behavior (Bongard-Blanchy et al., 2021; Mohammed et al., 2024). Week-long instructional units engaging ninth-graders in model construction and auditing demonstrate that students can successfully identify biased outputs and trace problems to their sources (Morales-Navarro, 2025).

Design research in high schools adds another layer of evidence. Game-based ethical-hacking curricula developed by Mohammed et al. (2024) demonstrate that even students in grades 9-12 with no prior experience quickly understand the legal-ethical boundary when courses frame red-teaming as a civic service rather than an act of rebellion.

These findings align with earlier work in white-hat training. Learners operating within normative frameworks that include public disclosure rules, reflection journals, and instructor oversight develop watchdog competence rather than illicit behavior tendencies. Withholding adversarial techniques doesn't preserve student virtue but instead leaves them defenseless. Teaching these techniques with appropriate ethical guardrails transforms potential vandals into the first line of collective safety.

# 4 Policy & Standards Agenda

## 4.1 Treat as a Curricular Mandate

The simplest lever for large-scale change is to treat adversarial reasoning as a required learning outcome, not an enrichment add-on. Two existing frameworks provide opportunities for immediate implementation. The Computer Science Teachers Association (CSTA) Standards already contain relevant elements that could be expanded. Strand 3B-AP-18 currently asks grades 11-12 students to "explain security issues that might lead to compromised computer programs" (CSTA, 2017). A straightforward revision adding "including adversarial testing of AI systems and disclosure of discovered vulnerabilities" would legitimize red-team practice immediately within existing educational structures.

Similarly, the College, Career, and Civic Life (C3) Social-Studies Framework includes components that align with adversarial reasoning education. Dimension 4, which focuses on Communicating Conclusions, emphasizes sourcing, credibility, and public argument (National Council for the Social Studies, 2013). Adding an exemplar such as "publish a public report documenting an AI model's bias or security flaw, with reproduction steps and mitigation proposals" would effectively connect technical critique to civic action within established educational guidelines.

Precedent for such additions already exists in state-level educational policy. Colorado's 2024 revised K-12 Computer Science Standards include an explicit Artificial Intelligence strand requiring high school students to "explain potential ethical dilemmas and biases in developing, training, and using AI tools" (CS.HS.5.1c) (Colorado Department of Education, 2024). Similarly, Texas' Career and Technical Education rules for Cybersecurity Capstone (19 TAC §127.770) mandate instruction on penetration testing fundamentals, including planning, authorization protocols, and vulnerability-scanning tools (Texas Education Agency, 2023). Expanding these existing standards from descriptive lessons to active adversarial testing would require only minor amendments or updated exemplar units without needing any new legislative framework.

Positioning adversarial reasoning within official standards simultaneously addresses both equity and safety concerns. When an activity becomes mandated, school districts must allocate funding for professional development and vetted sandbox platforms. This requirement ensures that low-resource schools receive the same ethically controlled learning environments as their affluent counterparts, as noted by Kafai & Burke (2014). Standardization therefore distributes both the skills and the ethical guardrails that transform red-team education from a private hobby into a genuine public good accessible to all students.

## 4.2 Extend the AI Bill of Rights to Youth

The White House Blueprint for an AI Bill of Rights (Office of Science and Technology Policy, 2022) asserts that every American deserves "explanation and control" when automated systems affect their lives. Students in Burriss et al. (2024)'s co-design study took this principle further. They demanded the ability to verify that AI explanations are truthful. As they put it, "Transparency without the power to test is performative." Translating this sentiment into policy means recognizing a Right to Red-Team—a protected entitlement for young people to probe, audit, and publicly report AI failures they encounter in school, social media, or consumer applications.

Codifying this right serves an important purpose. Without explicit legal protection, students who discover a prompt leak or bias pattern risk disciplinary action under computer-misuse statutes or school "acceptable-use" policies. Legal precedents already show that young security researchers can be punished even when they follow responsible-disclosure norms. For example, high-school student Bill Demirkapi was suspended after revealing critical vulnerabilities in his district's grading software, and Missouri Governor Mike Parson sought criminal charges against a reporter who quietly reported an exposure of 100,000 teacher Social Security numbers (Hancock, 2021). Establishing a formal right

would shield educational red-teaming with the same good-faith safe harbor that currently protects journalists and security researchers, as advocated by the Electronic Frontier Foundation (2023).

A model clause for this right might read:

> "Minors shall have the protected right, under supervised educational settings, to conduct adversarial tests on AI systems to evaluate bias, privacy, and security, provided they (a) obtain adult oversight, (b) limit testing to non-harmful payloads, and (c) publish findings through a responsible-disclosure channel."

This approach parallels the Coordinated Vulnerability Disclosure frameworks already adopted by major technology companies and federal agencies (CISA, 2024), but adapts them specifically for classroom oversight.

This Right to Red-Team would particularly benefit students in surveillance-heavy school districts, where algorithmic grading, facial-recognition discipline, or armed-violence prediction tools disproportionately affect marginalized youth, as documented by Benjamin (2019). Empowering these students to audit the systems that govern their educational experience is not merely sound pedagogy. It represents a fundamental civil rights issue, giving students agency within increasingly automated educational environments.

### 4.2.1 School-Level Responsible Disclosure Protocols

Implementing the Right to Red-Team requires clear procedures that protect students, teachers, and schools from legal liability. We propose a three-tier response framework adapted from the NTIA's coordinated vulnerability disclosure guidelines (NTIA, 2016) and CISA's implementation framework (CISA, 2024).

Students document low-severity findings (minor inconsistencies, easily corrected biases) and share them with their teacher as regular coursework. Teachers review these as learning artifacts. For medium-severity issues (reproducible jailbreaks, systematic biases, non-critical data exposures), teachers escalate to the school AI literacy coordinator within 24 hours. The coordinator verifies the finding and submits reports through vendor security channels while keeping student identities anonymous. For high-severity vulnerabilities (personal data exposure, safety risks, illegal activity facilitation), teachers immediately notify administration and the coordinator. Schools contact vendor security teams within 12 hours and may notify district IT security or relevant government agencies.

Legal protections require explicit safe harbor provisions. School acceptable use policies should include carve-outs for supervised educational red-teaming, mirroring protections for journalism students or science lab activities. Research on platform auditing shows that establishing formal channels for external scrutiny improves both system quality and public trust (Vaccaro et al., 2020; Brundage et al., 2020). Vendor agreements must permit supervised student testing and designate education liaisons for vulnerability reports. Experience from cybersecurity education programs shows these arrangements are feasible at scale (Air & Space Forces Association, 2024; Dark et al., 2021). State legislatures should extend educational immunity statutes to cover approved AI literacy activities, following frameworks established for other security research contexts (Buchanan & Benson, 2021; Applegate, 2014). School insurance policies should explicitly include this coverage.

Teachers determine whether student activity constitutes educational inquiry or potential misuse based on documented protocols. Did the student follow approved procedures? Report findings promptly? Limit testing to approved environments? Teachers use standard evaluation forms, with principals reviewing borderline cases. This parallels existing practices in ethical hacking education where clear protocols effectively guide student behavior (Hartley, 2015; Trabelsi & McCoey, 2016; Bongard-Blanchy et al., 2021). Students receive explicit training on responsible disclosure ethics and sign learning agreements establishing expectations before conducting adversarial activities, similar to safety protocols in chemistry or driver education.

## 4.3 Scaling Implementation Across Grade Levels

Adversarial reasoning can be taught at developmentally appropriate levels throughout K-12 education. Elementary students (K-5) learn to identify when AI-generated content might be incorrect using simple activities with image classifiers or chatbots that require no coding (Ali et al., 2019; Driscoll & Kumar, 2025). Middle school students (6-8) advance to guided adversarial prompting through workshop-based curricula. Three-hour sessions can shift students from naive trust to investigative skepticism (Solyst et al., 2024). High school students (9-12) engage in systematic red-teaming, auditing commercial models for bias and identifying failure modes (Morales-Navarro, 2025).

Teacher preparation follows a tiered model. Initial professional development requires 6-12 hours covering basic AI literacy, facilitation of adversarial reasoning activities using prepared lesson plans, and ethical frameworks for responsible disclosure. This training can be delivered through weekend workshops or online micro-credentials (McGill et al., 2022). Ongoing support includes access to curated lesson libraries, secure sandbox platforms, and peer learning communities.

Assessment focuses on process rather than outcomes. Students demonstrate competency by documenting their testing methodology, explaining their reasoning, and articulating ethical considerations. Rubrics evaluate whether students can formulate testable hypotheses, interpret model responses, identify patterns in failures, and propose responsible next steps.

Safe sandbox environments must include technical safeguards. Rate limiting prevents students from overwhelming systems. Content filtering blocks attempts to generate harmful material. Audit logging records all queries for teacher review. Sandboxes should use isolated instances or API keys with restricted permissions. State and federal grants should prioritize funding for these infrastructure costs to ensure equitable access (Federal Emergency Management Agency, 2024; National Science Foundation, 2024).

## 4.4 Equitable Funding and Infrastructure

Curriculum mandates without resources simply widen the digital divide. A rural district lacking GPUs or dedicated computer science teachers cannot create a safe red-team sandbox. History shows that unfunded technology mandates systematically disadvantage low-income schools, as documented by Margolis et al. (2008). We therefore propose a National Adversarial-Reasoning Grant Line modeled on two proven funding approaches.

Since 2012, the Cybersecurity Education and Training Assistance Program (CETAP) has been CISA's main channel for delivering cybersecurity curriculum and professional development to K–12 schools. In fiscal year 2023, CISA awarded 6.8 million dollars through CETAP, support that activated more than 30,000 educators across every state and four U.S. territories on the CYBER.ORG platform (Cybersecurity and Infrastructure Security Agency, 2024). An independent evaluation of a 30-school CETAP cohort reported that participating Title I schools expanded computer-science and cybersecurity course offerings more rapidly than comparable non-Title I schools (McGill et al., 2022).

The National Science Foundation's Computer Science for All Researcher-Practitioner Partnerships program provides a complementary scale-up path. The current solicitation allows Small RPP awards of up to 300 thousand dollars for two-year pilots and Medium RPP awards of up to one million dollars for three-year, multi-district projects (National Science Foundation, 2024). Initiatives that include white-hat hacking experiences, such as GenCyber Girls and Cyber Up!, have documented significant gains in female students' cybersecurity self-efficacy and a higher likelihood that participants remain on computing pathways (Dark et al., 2021; Podhradsky et al., 2018). These outcomes mirror national Advanced Placement trends showing that between 2017 and 2020 the share of women and students of color taking the AP Computer Science Principles exam more than doubled (College Board, 2020).

Allocating just $25 million a year, roughly 0.8 percent of the $3.1 billion the Federal Government requested for core AI R&D in FY 2024 (National Science and Technology Council, 2023), would let every U.S. public high school stand up at least one secure sandbox and train one instructor. States could meet the other share by adopting the 70/30 (or 80/20 for regional consortia) match already embedded in DHS's State and Local Cybersecurity Grant Program (Federal Emergency Management Agency, 2024), which Governors have endorsed as a workable model. Linking awards to open-reporting would turn classrooms into a nation-wide early-warning network instead of isolated experiments.

# 5    Conclusion

Large-scale language models now mediate search, homework help, college essays, customer service, and political mobilization. Their very design, combining massive pre-training plus ever-shifting system prompts, means that *perfect, static alignment is mathematically out of reach* (Glukhov et al., 2023). The sycophancy bug that slipped through OpenAI's GPT-4o release and the conspiracy prompt leaked by Grok are therefore not embarrassing footnotes; they are early warnings of a structural reality: as capabilities climb, so will novel failure modes and adversarial exploits. Relying on closed, vendor-run red teams to spot every breach is wishful thinking. Adaptive suffixes (Andriushchenko et al., 2025), ASCII payloads (Jiang et al., 2024), and prompt-leak frameworks (Hui et al., 2024) keep proving otherwise.

Our position paper has advanced three claims. **First**, civic accountability requires that the public possess not merely *critical-reading* skills but *critical-probing skills*, what we call adversarial reasoning. This follows directly from children's rights to explainability and agency in systems that govern them (Adams et al., 2023) and from democratic theory's demand for contestability of power. **Second**, adversarial reasoning is *already teachable*: week-long or even 3-hour modules shift middle-schoolers from naïve trust to investigative skepticism without fostering misuse (Solyst et al., 2024; Morales-Navarro, 2025; Bongard-Blanchy et al., 2021). **Third**, embedding this competence at scale is practical: a one-line addition to CSTA and C3 frameworks, a "Right to Red-Team" clause in youth AI Bills of Rights (Burriss et al., 2024), and a $25 million federal grant line, 0.8% of current AI-research outlays, would provide every U.S. public high school a safe sandbox and a trained facilitator.

Critics fear that teaching offense will mint hackers, yet the data run the other way: guided red-team courses correlate with *lower* malicious behavior and higher rates of responsible disclosure (Bongard-Blanchy et al., 2021). Others argue we should simply "align harder." But alignment remains a moving target; educating watchdog citizens is the only solution that scales with the problem. Finally, some invoke teacher capacity. Micro-credential models from cybersecurity show that modest stipends and asynchronous professional development are enough to bring non-specialists on board, particularly when curricula center on puzzles and civic debate rather than deep code.

In 1955, educators added atomic-age "duck-and-cover" drills; in 1985, they folded HIV awareness into health class; in 2005, they adopted phishing simulations in computer labs. In 2025, the analogous civic upgrade is adversarial reasoning. A generation that can stress-test the systems shaping its future is a generation equipped to steer that future, to call out hidden biases, disclose prompt leaks, and insist on safer, more accountable AI. The choice before policymakers and curriculum designers is stark: cultivate watchdogs now or confront avoidable failures later. The cost of inaction will extend far beyond buggy chatbots to affect public trust, democratic resilience, and the epistemic health of the next electorate (Dabbagh et al., 2025). We urge standards bodies, legislators, and funders to act, because when a single unseen prompt can distort billions of conversations overnight, *passive literacy is no longer literacy enough*.

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
