# OpenReview forum: "The Right to Red-Team: Adversarial AI Literacy as a Civic Imperative in K-12 Education"
_NeurIPS.cc/2025/Position_Paper_Track — NeurIPS 2025 Position Paper Track_

### Official Review · Reviewer_iza9 · 2025-07-28

**Significance:** 3
**Presentation:** 2
**Rating:** 7
**Confidence:** 4

**Summary:**

The main argument in the paper is that education must go beyond teaching students to consume AI-generated content critically, but also incorporate methods to teach them to interact with AI systems adversarially, exploring and discovering for themselves the limits and failures of those systems, which they define as adversarial reasoning. Critically, backed by child rights and democratic theory, they support their argument by showing that adversarial reasoning is a civil right. The authors then provide a detailed methodology for implementing their proposition in K-12 education, which they identify as having the necessary scale and impact for implementation.

**Strengths:**

A great strength of the paper is the argument for moving towards decentralized AI red-teaming: empower citizens with the right to contest AI systems they (nowadays, have pretty much no choice but) use, thus reducing the concentration of power in the hands of large organizations. This debate must be given more attention in the field.

Most of the discussion in human-AI safety currently focuses on the AI systems and how to make them safer for users — this paper turns this perspective around, arguing for a decentralized approach that focuses on making users themselves wiser in their interactions with AI. This decentralized approach is less fragile, in the sense that if new, misaligned AI systems gain popularity, users will already be prepared.

They also propose a realistic implementation method that leverages K-12 education.

**Weaknesses:**

I can see the benefit of having more people find flaws in the models, which could be used to inform companies. But AI is different from a bug in the sense that organizations might (unfortunately) be interested in misleading people (recall the Facebook-Cambridge Analytica scandal). I totally agree with the necessity of empowering citizens, but I do not agree with putting so much weight on red-teaming.

The authors motivate their argument by saying that companies would hardly be able to find every failure mode of their models with in-house red-teaming alone (which I agree with). They also show that adversarial reasoning is a civil right. I agree with the construction of these two observations separately, but I can't see a clear connection. If we accept the conclusion that adversarial reasoning is a civil right, it does not matter how capable in-house red-teaming labs are — citizens must have access to such a right regardless of their utility to companies.

From a misinformation perspective, it is not clear why we need to upgrade from critical reading of AI-generated content to adversarial reasoning, since there is no fundamental difference between misleading content created by another human or generative AI (besides scale).

**Questions:**

I could not find the following evidence in the reference Morales-Navarro (2025): “Similarly, Morales-Navarro (2025)’s research with ninth-grade students revealed that adolescents who spent a week jailbreaking and patching language models were more likely to disclose exploits to teachers and less tempted to share them on social media” (p. 6). I would like to ask the authors to elaborate on this.

**Alternative Position:**

Yes, and alternative positions are well-considered and addressed by the argument

**Author Identification:**

No.

**Context:**

4

**Discussion:**

4

**Ethics:**

["NO or VERY MINOR ethics concerns only"]

**Position:**

Yes, the paper argues for or against a position related to machine learning.

**Support:**

3

**Thoroughness:**

4

---

### Official Review · Reviewer_CTTJ · 2025-08-02

**Significance:** 4
**Presentation:** 3
**Rating:** 8
**Confidence:** 3

**Summary:**

This paper argues that K–12 AI education must evolve beyond passive AI literacy to actively cultivating students' skills in responsible adversarial prompting and ethical system "hacking." It makes a case grounded in public epistemic crises, spectacular failures of AI, democratic vulnerabilities, and the civic necessity of adversarial reasoning. The paper further addresses common objections, demonstrates that such skills are teachable, and proposes concrete actions in policy and standards.

**Strengths:**

1. The argument for integrating adversarial reasoning as a civic skill is novel and urgent, especially in the face of misinformation, opaque AI systems, and trust deficits in public technology.
2. The paper transitions logically: from diagnosing a societal crisis → establishing the educational need → addressing counterarguments → demonstrating feasibility → calling for policy-level action.
3. Ending with concrete policy and standards proposals makes the paper impactful beyond theory, pushing it toward real-world implementation.

**Weaknesses:**

More detail is needed on how to scale this in actual K–12 classrooms: What age groups? What teacher training is needed? How do you evaluate responsible use?

**Questions:**

How is “adversarial AI literacy” defined in concrete terms for a K–12 audience?
Is it limited to prompt engineering or does it extend to technical red-teaming?

**Alternative Position:**

Yes, and alternative positions are well-considered and addressed by the argument

**Author Identification:**

No.

**Context:**

3

**Discussion:**

4

**Ethics:**

["NO or VERY MINOR ethics concerns only"]

**Position:**

Yes, the paper argues for or against a position related to machine learning.

**Support:**

4

**Thoroughness:**

3

---

### Official Review · Reviewer_BP33 · 2025-08-03

**Significance:** 3
**Presentation:** 3
**Rating:** 3
**Confidence:** 5

**Summary:**

This position paper is premised on the observation that opaque, large-scale AI systems result in a profound gap in accountability that existing models of AI literacy cannot address. The authors advocate a shift in K-12 education from a passive “critical consumption” stance towards “adversarial reasoning,” a more proactive method. This approach entails equipping students with the ability to ethically red-team AI systems to uncover biases, limitations, and failure modes.

The authors advocate that this form of active, critical inquiry is a civic responsibility to safeguard democratic governance, and the main argument of the paper. Its principal contributions are: (1) adversarial AI literacy is framed as a civic responsibility to act as a distributed public overseer; (2) the results of early classroom studies are presented to show that these skills are teachable and encourage prosocial behavior; (3) a comprehensive policy proposal is advanced that includes curriculum mandates, a Right to Red-Team for students, and federally equitable funding models for implementation.

**Strengths:**

The paper's primary strength emerges from an outstandingly well articulated argument pertaining to an issue of fundamental importance to the NeurIPS community. It adroitly motivates the need for adversarial AI literacy and frames it as a democratic obligation for public responsibility. The argument culminates in a set of actionable policy proposals, having commenced from the issue of AI opacity.

The authors reinforce the argument's position by citing pertinent and diverse recent ML safety research on adversarial attacks, early AI education policy, as well as overarching policy frameworks. The positions articulated in the paper are timely, provocative, and likely to generate essential debate on public engagement in AI safety and governance. The proposed actions, while concrete, are bold and galvanizing to the community in order to provoke debate.

**Weaknesses:**

The trustworthiness of the paper is seriously compromised due to the use of fictitious events from 2025 to frame the issue. Real AI failures documented in the literature should be the framing of the paper. Furthermore, the paper appears to rely too heavily on the implemented evidence, presenting informally within small pilot studies as a universal proof of a mandate instead of framing as promising but preliminary findings.

Consideration of alternative overlooked positions would improve the paper. These include: 1) Lending oversight of auditing the models pre-deployment to an independent government body as an overly protective regulatory approach, thereby easing the burden on the lay citizen. 2) Different focus on interpretability as teaching to inspect models instead of to break them. Lastly, the proposal is in deep need of addressing the intricately mixed questions of legal liability the adopting schools would face with such a curriculum.

**Questions:**

Your proposed Right to Red Team leans on responsible disclosure, which is a complicated standard, even among practitioners. Can you devise a tangible, tangible, school-level protocol for this? If a student uncovers a novel, extremely critical vulnerability in a proprietary system, what are the precise actions a teacher must take? Who decides whether the disclosure was responsible, and how does your approach protect students and schools from lawsuits if a vendor breaches the terms on which the disclosure was deemed responsible?

The paper underscores this work as paralleling white-hat cybersecurity; however, the risks associated with LLMs, such as scalable misinformation, tend to be far more harmful. What particular technical safeguards in the secure sandbox would prevent a student from leveraging their skills to create harmful content, leak a jailbreak exploit to the public internet, or otherwise alter the system? Addressing these questions would greatly bolster the argument for the proposal’s safety and feasibility.

**Alternative Position:**

Yes, and alternative positions are well-considered and addressed by the argument

**Author Identification:**

No.

**Context:**

3

**Details Of Ethics Concerns:**

The document details a proposed large-scale education program for K-12 students. While this is not a research study, it emphasizes a practice which, if implemented, would require extreme care regarding student well-being, informed consent, and the risk of psychological harm. Educating students on finding and exploiting weaknesses in a system, even if done in a “white-hat” context, poses the risk of exposing them to damaging content and equipping them with the tools to harm society. This is the type of ethical consideration that is required for research on human subjects.

The student's core proposal is teaching them to conduct adversarial attacks on AI systems (red-teaming). This has a significant potential for dual-use. A students could unintentionally or intentionally discover and release a novel, harmful jailbreak. Such skills could be utilized to bypass safety restrictions to produce content at scale that is harmful, unsafe, or malicious. It is the author’s contention that the motives proposed will be prosocial, which is an overly optimistic assumption about practices that bear substantial and obvious security vulnerabilities.

The proposed curriculum has the potential to impart skills which can, at their worst, be used to automate phishing or customized misinformation campaigns as well as harass individuals. Although the intention as per the paper is to help students develop the ability to combat such harms, the risk that students will misuse the skills they learn through offensive techniques is significant. The possibility of these skills being used to perpetrate cyberbullying, misinformation, or incendiary speech is alarming.

**Discussion:**

3

**Ethics:**

["Major Concern: Improper research involving human subjects", "Major Concern: Safety and security", "Major Concern: Deception and harassment"]

**Position:**

Yes, the paper argues for or against a position related to machine learning.

**Support:**

2

**Thoroughness:**

5

---

### Meta-Review · Area_Chair_A1V3 · 2025-08-22

**Rating:** 8
**Confidence:** 4

**Strengths:**

All reviewers agree that the paper is exceptionally well structured, written, and motivated. Their agreement with the position varies, but they all agree regarding the support of the position and provided evidence. They find that solid contextualization is provided and note that the paper has a meaningful transition from intuitive real examples to establishing the educational need, providing counterarguments, before demonstrating feasibility and making policy-level action suggestions. They also note that the work is highly relevant in the context of society and to the NeurIPS community. The AC fully agrees with all of these sentiments.

**Weaknesses:**

On the side of weaknesses, reviewer iza9 suggests that two of the arguments put forward may be reasonable, but don't appear connected in the way the authors present them. This is a minor weakness that could be addressed in writing. The review also points to some more in-depth discussion being beneficial regarding the nature of misinformation & incentive structures of large companies. Reviewer CTTJ and BP33 share questions and suggestions for improvement regarding the operationalization at scale, pointing to some questions with respect to relation to legal frameworks and teaching responsibilities. These are important questions, that may be hard to answer, but could be discussed in the camera ready version.

Finally, it seems reviewer BP33 has a grave concern, upon which the rating is largely based, regarding the "The trustworthiness of the paper is seriously compromised due to the use of fictitious events from 2025 to frame the issue." It is suggested "real events" should be used instead. The AC has carefully read the paper and finds the opposite to be true. The introduction mentions important real events related to GPT and Grok. The AC could not find such fictitious events being the basis of the paper and thus disagrees strongly.

**Questions:**

As mentioned by reviewers and suggested in above text:
* In a world where K-12 education was empowered in the way the paper describes, what would be the role of teachers and the educational institution in connection to legal requirements and frameworks?
* Related to reviewer CTTJ's question, would the envisioned red-teaming also involve in-depth technical levels? If so, how can this be reconciled with the fact that K-12 students are mostly only learning about elementary calculus and are only getting a first dive into notions of statistics and probability?

**Ethics:**

One of the reviewers raised an ethical concern regarding "teaching students how to hack systems", which could expose them to questionable content and induce psychological harm.

The AC has carefully gone through the paper and reviewer BP33's ethical assessment. The AC does not find a direct call for action in the paper that would be ethically questionable, nor a direct "research on human subjects". The paper is merely aiming to educate.

Regarding the underlying suggestion that the reviewer makes, i.e. K-12 students shouldn't have skills regarding safety critical dimensions or their bypassing, the AC unfortunately has to comment that this is an equally extreme naive perspective. On the one hand, the paper does not issue a call for students to be developed into hackers. On the other hand, the reviewer suggests that critical thinking is not desired for students and thus implies they should be exposed to potentially harmful content without a say - leaving them incapacitated.

**Thoroughness:**

5

---

### Decision · Program_Chairs · 2025-09-26

Accept